# Dynamic Characteristic Master Curve and Parameters of Different Asphalt Mixtures

**Shijie Ma [1,2], Liang Fan [1,*], Tao Ma [3], Zhao Dong [1], Yuzhen Zhang [4] and Xiaomeng Zhang [1,*]**

1   Shandong Transportation Institute, Jinan 250012, China; mashijie@sdjtky.cn (S.M.); dongzhao@sdjtky.cn (Z.D.)
2   School of Transportation, Shandong Jianzhu University, Jinan 250101, China
3   School of Transportation, Southeast University, Nanjing 211189, China; matao@seu.edu.cn
4   College of Chemistry and Chemical Engineering, China University of Petroleum, Qingdao 266580, China; zhangyuzhen@sdjtky.cn
*   Correspondence: fanliang@sdjtky.cn (L.F.); zhangxiaomeng@sdjtky.cn (X.Z.)

**Abstract:** Using an AMPT tester and based on laboratory tests, this paper performed a comparative study on the dynamic characteristics of different asphalt mixtures, analyzed the influence of different asphalt binders on the characteristic parameters of the dynamic modulus master curve and the phase angle master curve of asphalt mixture, and expounds the evaluation function of the phase angle master curve for mixture relaxation characteristics. The results show that the modulus master curve parameters of the asphalt mixture are closely related to voids in the mineral aggregate, mixture density, and asphalt content of the asphalt mixture. For the same kind of asphalt mixture, because the gradation of mineral aggregate is fixed and the volume parameters are almost the same, the ultimate modulus of the mixture at different temperatures is unique; when the temperature changes or the asphalt changes, the shape parameter β of the modulus master curve changes regularly, which brings different dynamic responses, and the lower β will show the characteristics of a higher modulus. Asphalt is the source of the viscoelasticity of the asphalt mixture. Although the influence of particle gradation of the mixture will bring about the change of modulus, the phase angle of the mixture depends on the viscoelastic properties of asphalt, and the initial phase angle in the main curve is positively correlated with asphalt penetration and negatively correlated with the softening point and viscosity, while the peak phase angle A is negatively correlated with penetration, and the softening point viscosity is positively correlated. The viscoelastic interval, represented by ω, is negatively correlated with penetration but positively correlated with the softening point and viscosity. The peak position, parameter ωc, of the phase angle master curve can evaluate the relaxation characteristics of the mixture, and the crack resistance of different mixtures can be compared without complex model calculation. In the comparison of the relaxation time of asphalt mixture, the relaxation time of foam cold-recycled mixture is the largest, which is significantly higher than that of other forms of cement mixture; the emulsified asphalt cold-recycled mixture is equivalent to AC20 and LSPM30 mixtures; the SBS-modified asphalt mixture has the best relaxation characteristics.

**Keywords:** asphalt mixture; master curve; characteristic parameter; relaxation characteristics; cold recycling; foamed asphalt; emulsified asphalt

## 1. Introduction

In the 1970s, the dynamic performance master curve was first used to study polymer damping materials and was used as the main content of the product specification of damping materials [1]. The master curve contains the basic dynamic parameters of viscoelastic materials, including dynamic modulus and phase angle, both of which express the material properties as a whole, and the phase angle can better express the relaxation characteristics of the material. Materials with the same dynamic modulus and different phase angles are different materials [2,3]. However, in the research and application of master curves, it is

not appropriate to pay more attention to dynamic modulus and ignore the study of phase angle master curves.

At present, the dynamic performance of the asphalt mixture is also characterized by the method of the master curve [4–6]. The actual state of the asphalt pavement under dynamic load is complex, which can be regarded as the superposition of multiple sine waves. The response of asphalt pavement materials to a single sine wave is the basic dynamic property. Because asphalt pavement material is a viscoelastic system, its dynamic modulus and phase angle parameters constitute the basis for understanding pavement load response [7]. Generally speaking, the mechanical response of asphalt pavement is closely related to driving speed; that is, the stiffness modulus is dependent on the loading time, and the shorter the stress pulse time is, the greater the modulus of the asphalt mixture is, which has the characteristics of high frequency and high elasticity [8–18]. However, the current research pays more attention to the dynamic modulus of asphalt pavement and does not perform an in-depth study on the phase angle, which is an important parameter reflecting the viscoelastic structure. Based on laboratory tests, this paper studied the dynamic response characteristics of different asphalt mixtures and discussed the influence of the asphalt binder on the characteristic parameters of the dynamic modulus master curve and phase angle master curve of the asphalt mixture. The evaluation functions of phase angle master curve and mixture relaxation characteristics are expounded.

## 2. Acquisition of Dynamic Principal Curve

The asphalt mixture was tested by an AMPT performance tester, and the modulus and phase angle at 20, 30, 40, and 50 °C under nine frequency conditions in the range of 0.1 to 25 Hz were obtained, and the modulus and phase angle at different temperatures were transformed into master curves at the same reference temperature by using the principle of time–temperature equivalence. Based on the existing research, the Sigmoid model was used to fit the dynamic modulus master curve, and the improved Gauss model was used to fit the phase angle master curve [9,19].

(1) Dynamic modulus master curve

The Sigmoid model of the modulus master curve is shown in Equation (1).

$$lg(E^*) = E_0 + \frac{\alpha}{1 + e^{\beta + \gamma(lg\omega_r)}} \tag{1}$$

Among them, E* is the dynamic modulus, $\omega_r$ is the load frequency at the reference temperature, $E_0$ represents the minimum value of dynamic modulus, $E_0 + \alpha$ represents the maximum value of dynamic modulus, and β and γ are the shape parameters of the model curve.

(2) Phase angle master curve

The phase angle master curve was fitted according to the improved Gauss model with the same displacement factor as the modulus principal curve. The Gauss model is shown in Equation (2).

$$\delta = \delta_0 + Ae^{\frac{(lg\omega_r - \omega_c)^2}{2\omega^2}} \tag{2}$$

In the formula, δ is the phase angle, $\omega_r$ is the frequency at the reference temperature, $\omega_c$ is the frequency corresponding to the peak of the phase angle, and $\delta_0$, A is the eigenvalues of the master curves.

Figure 1 shows the master curve of the AC-20 type mixture. Among them, the modulus master curve in Figure 1A shows a smooth S-shaped curve, which reflects the mechanical properties of the full frequency range. The asphalt mixture has the characteristics of high frequency and high elasticity, and the dynamic modulus increases and tends to have a stable value at high frequency; under the condition of low frequency, the dynamic modulus decreases and tends to a minimum.

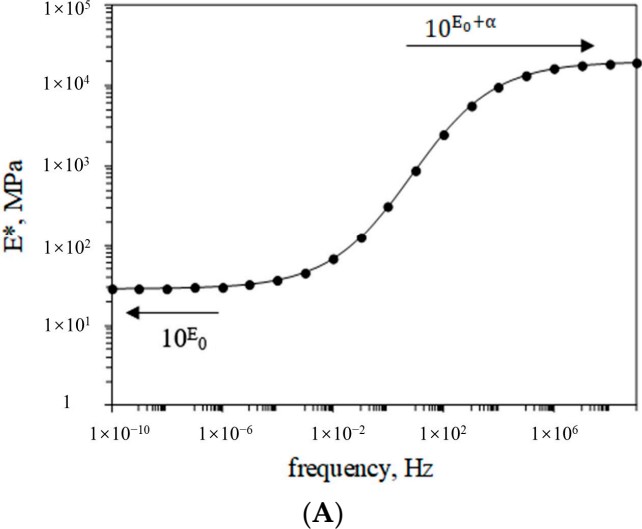

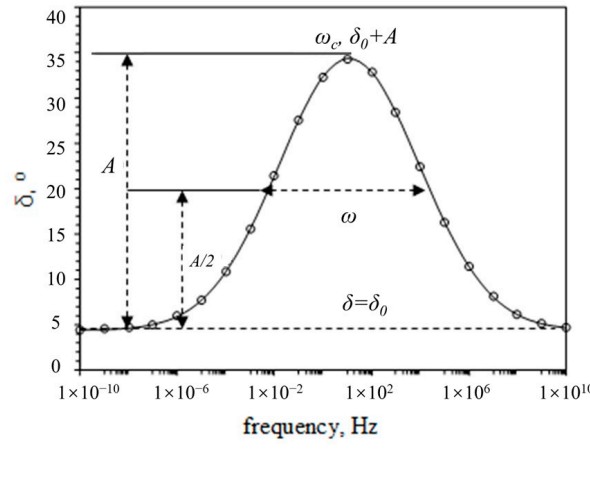

**(A)**　　　　　　　　　　　　　　　　　　　　　**(B)**

**Figure 1.** Example of master curves of the asphalt mixture. (**A**) Dynamic modulus master curve, (**B**) Phase angle master curve.

Figure 1B is the phase angle master curve. It can be seen that parameter $\delta_0$ represents the initial phase angle, parameter A represents the variation amplitude of the phase angle, parameter $\omega$ represents the full width at half maxima (FWHM) of the phase angle curve, and $\omega_c$ represents the position of the phase peak. According to the theory of polymer physics, the FWHM parameter of the phase angle master curve covers more of the viscoelastic state of the material, the left range tends to the rubber state, and the right range tends to the glass state [10,20,21]. In this paper, the division between the amplitude of phase angle A and FWHM ($A/\omega$) is used to represent the significant degree of viscoelasticity of the material; the higher the $A/\omega$ value, the greater the sensitivity of the material to frequency, and vice versa.

## 3. Characteristic Parameters of Master Curves

Taking the same hot-mix asphalt mixture AC-20 as the research object, three kinds of asphalt mixtures were prepared according to different petroleum asphalts. The volume index of the mixture is shown in Table 1. The dynamic modulus and phase angle of the mixture were obtained on the AMPT tester, and the main curve was drawn by model fitting according to Equations (1) and (2). The characteristic parameters of the main curve are shown in Table 2, and the master curves are shown in Figure 2.

**Table 1.** Asphalt and asphalt mixture indexes.

| Mixture Type | ZH50-AC20 | ZH70-AC20 | ZH90-AC20 |
|---|---|---|---|
| Penetration grade | 50 | 70 | 90 |
| Asphalt content (Pb) % | 4.3 | 4.4 | 4.3 |
| Mixture density ($\gamma_f$) g/cm$^3$ | 2.42436 | 2.42498 | 2.4296 |
| Mixture void percent (Vv) % | 4.59 | 4.42 | 4.4 |
| Aggregate void ratio (VMA) % | 14.15 | 14.1 | 14.4 |
| Asphalt saturation (VFA) % | 67.32 | 68.54 | 68.47 |

**Table 2.** Master curve parameters for three asphalt mixtures.

| Type | Temperature, °C | Fitting Parameters for E* Master Curve | | | | | Fitting Parameters for δ Master Curve | | | | |
|---|---|---|---|---|---|---|---|---|---|---|---|
| | | $E_0$ | $\alpha$ | $\beta$ | $\gamma$ | $R^2$ | $\delta_0$ | $A$ | $\omega_c$ | $\omega$ | $R^2$ |
| ZH50-AC20 | 20 | 1.4522 | 2.844 | −1.537 | −0.664 | 0.997 | 4.436 | 29.961 | −2.084 | 2.893 | 0.989 |
| | 30 | | | −0.743 | | | | | −0.889 | | |
| | 40 | | | −0.046 | | | | | 0.161 | | |
| | 50 | | | 0.561 | | | | | 1.074 | | |
| ZH70-AC20 | 20 | 1.4926 | 2.7866 | −1.339 | −0.694 | 0.995 | 15.259 | 19.253 | −1.607 | 2.063 | 0.990 |
| | 30 | | | −0.607 | | | | | −0.551 | | |
| | 40 | | | 0.149 | | | | | 0.536 | | |
| | 50 | | | 0.924 | | | | | 1.654 | | |
| ZH90-AC20 | 20 | 1.3762 | 2.9551 | −0.998 | −0.627 | 0.996 | 18.223 | 16.309 | −1.216 | 2.025 | 0.987 |
| | 30 | | | −0.461 | | | | | −0.364 | | |
| | 40 | | | 0.156 | | | | | 0.624 | | |
| | 50 | | | 0.846 | | | | | 1.725 | | |

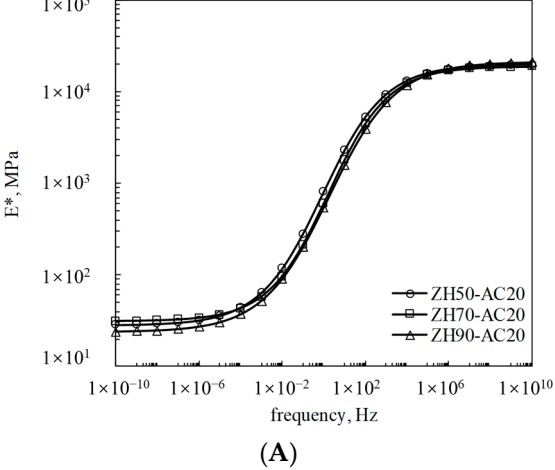

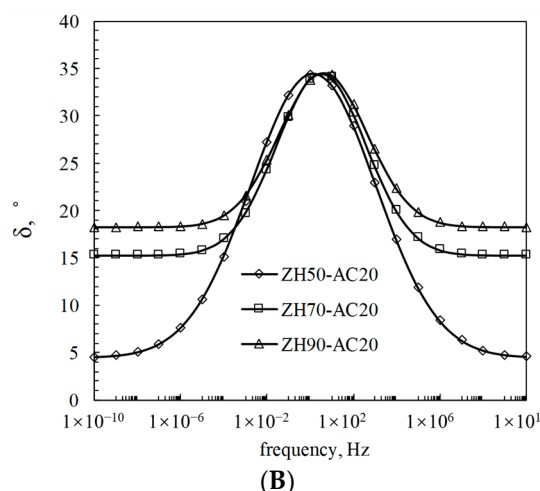

**(A)**            **(B)**

**Figure 2.** Master curves of three kinds of asphalt mixtures. (**A**) Dynamic modulus master curve, (**B**) Phase angle master curve.

### 3.1. Modulus Master Curve Parameters

Figure 2A shows that the modulus master curve of the asphalt mixture tends to a stable modulus value—the minimum modulus value ($10^{E_0}$) and the maximum modulus value ($10^{E_0+\alpha}$) under the condition of extremely low frequency and extremely high frequency. This shows that the dynamic modulus of the asphalt mixture at very low frequency (or high temperature) and very high frequency (low temperature) is not affected by load frequency and only shows obvious frequency sensitivity in the general range of $1 \times 10^{-5}$ to $1 \times 10^5$ Hz. According to the related research, the ultimate modulus of the asphalt mixture is only related to the mixture gradation, which depends on the mixture volume parameters and presents uniqueness [9,11]. Due to the use of the same gradation, the volume indexes of the three kinds of asphalt mixtures are almost the same. The ultimate modulus is also close to each other, and the difference is small. The parameters $\alpha$ and $\gamma$ of the principal curve of modulus are also stable.

The linear correlation analysis was carried out by using the master curve parameters of three kinds of mixture modulus ($E_0$, $\alpha$, $\gamma$) and the volume index of the mixture. Figure 3 shows that the correlation coefficients of the aggregate void ratio (VMA), mixture density ($\gamma_f$), and asphalt content (Pb%) are the highest, while the effects of the asphalt saturation (VFA) and void percentage (Vv%) are not significant.

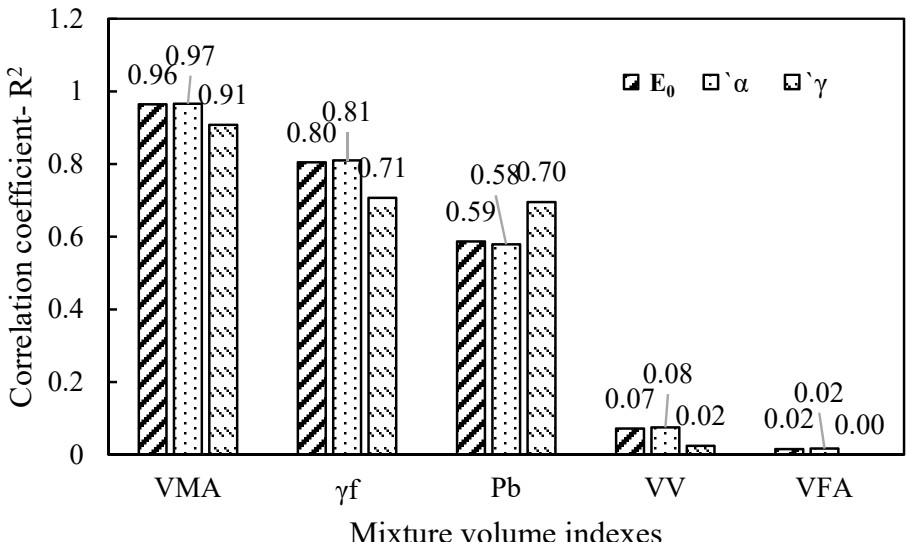

**Figure 3.** Correlation between the modulus master curve parameters and mixture volume indexes.

However, the parameter β of the master curve increases monotonously with the increase of the reference temperature and presents the linear relationship shown in Figure 4. This phenomenon is related to the shift factor of the principal curve and is essentially related to the temperature sensitivity of asphalt. A good power-law relationship can be established by using the principal curve parameter β at 20, 30, 40, and 50 °C and the complex modulus (G*) of asphalt at the same temperature (test conditions: stress 100 Pa, frequency 5 rad/s). Figure 5 shows that the shape parameter β of the main curve of mixture modulus decreases with the increase of complex modulus G* of asphalt (when the temperature decreases or the asphalt hardens). Under the same gradation condition, a low β value is more likely to lead to a higher modulus. In the range of $10^{-5}$ to $10^5$ Hz, the modulus of mixture made by penetration grade 50 asphalt is slightly higher than that of penetration grade 70 and grade 90 asphalt. Relative to the influence of modulus brought by gradation, the influence of asphalt is weak.

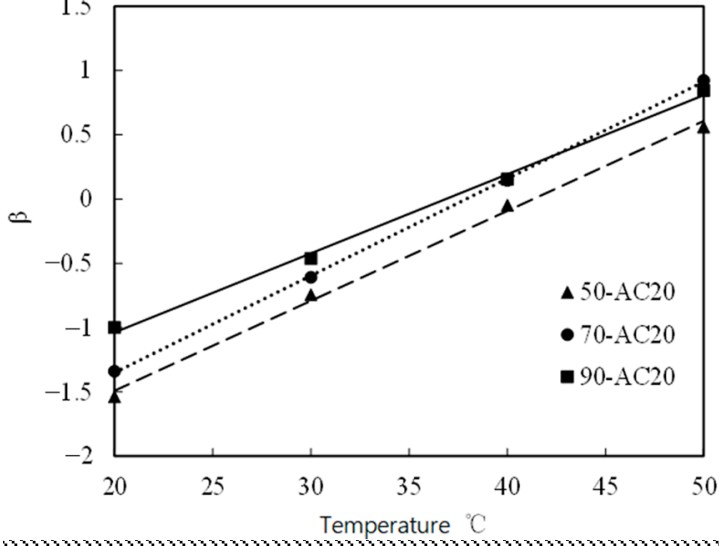

**Figure 4.** Relationship between shape parameter β and temperature.

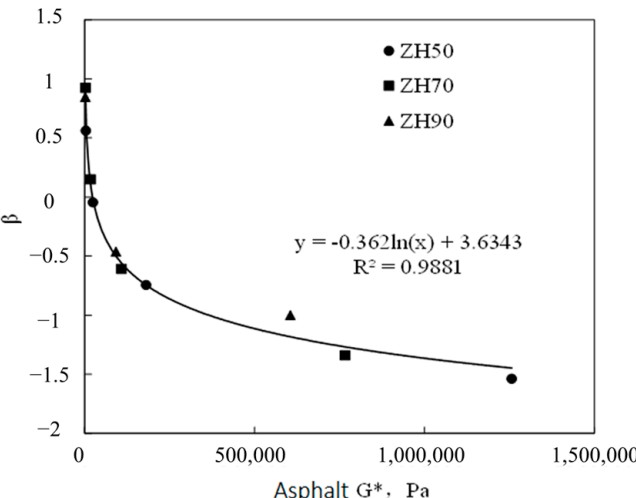

**Figure 5.** Relationship between asphalt modulus and shape parameter β.

It can be said that for the same kind of asphalt mixture, due to the fixed gradation of mineral aggregate and little difference in volume parameters, the ultimate modulus of the mixture at different temperatures is unique; when the temperature changes or the asphalt changes, the shape parameter β of the modulus master curve changes regularly, which brings different dynamic responses, and the lower β will show the characteristics of higher modulus.

### 3.2. Phase Angle Master Curve Parameters

#### 3.2.1. $\delta_0$, A, and $\omega$

In the same mixture, different asphalt will create a difference in the viscoelasticity of the mixture, which is mainly reflected in the change of the master curve parameters of the phase angle. Table 1 and Figure 2B show that the phase angle master curve parameter $\delta_0$ increases significantly with the increase of asphalt penetration grade, and the height parameter A of the master curve decreases with the increase of asphalt penetration grade. The half-width parameter $\omega$ decreases with the increase of asphalt penetration grade. This shows that different asphalt brings different viscoelastic characteristics to the same mixture; the viscous component of high-grade asphalt is large, so the viscous component of asphalt mixture is also high, resulting in a higher initial phase angle ($\delta_0$), which brings a narrower phase angle change range (A value) and a narrow viscoelastic response frequency range, which is reflected in the reduction of FWHM value ($\omega$).

Figure 6 shows the correlation analysis between the phase angle curve parameters and the asphalt indexes. The master curve of the phase angle of the mixture is closely related to the asphalt properties. The initial phase angle ($\delta_0$) is positively correlated with penetration and negatively correlated with softening point and viscosity; phase angle height A is negatively correlated with penetration; softening point viscosity is positively correlated; and the viscoelastic interval, represented by FWHM ($\omega$), is negatively correlated with penetration, but positively correlated with softening point and viscosity. It can be considered that asphalt is the source of the viscoelasticity of the asphalt mixture. Although the influence of particle gradation of the mixture will bring about the change of the modulus, the phase angle of mixture depends on the viscoelastic properties of asphalt, which is consistent with the previous studies; that is, there is a good sinusoidal relationship between the phase angle of asphalt and the asphalt mixture [11,22–24]. Moreover, the phase angles of different asphalt mixtures from previous studies, stone mastic asphalt (SMA-13), large-stone porous asphalt mixture (LSPM-25), F fine asphalt mixture AC-13F, and asphalt mixture AC-25, are shown in Figure 7.

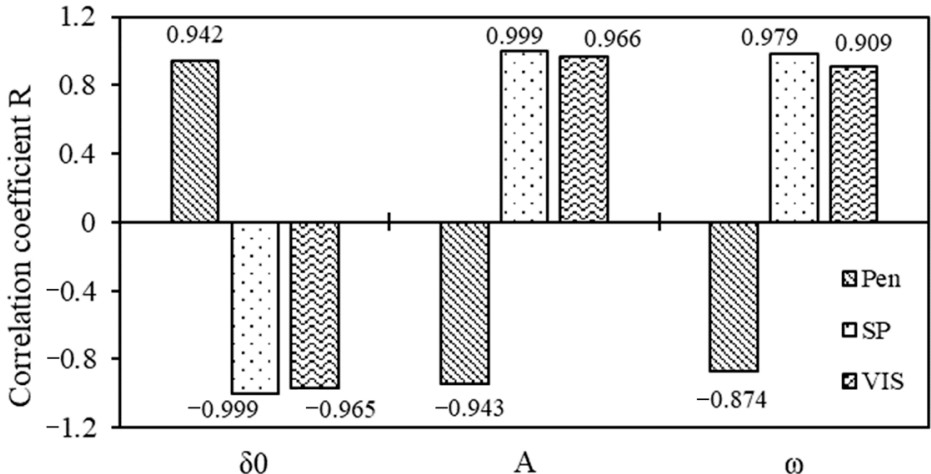

**Figure 6.** Correlation between master curve parameters of phase angle and asphalt indexes.

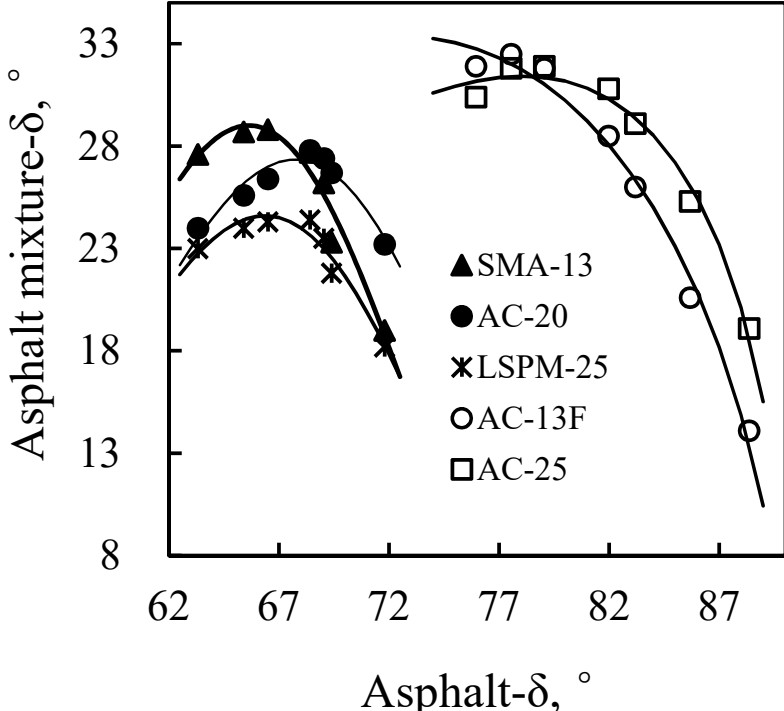

**Figure 7.** Phase angle relationship between asphalt and mixture.

3.2.2. Phase Angle Peak Position Parameter-$\omega_c$

$\omega_c$ is a parameter that varies with temperature. Figure 8 shows that the $\omega_c$ value of the phase angle master curve moves to the high-frequency position with the increase of temperature, which shows that the viscoelasticity of the asphalt mixture changes with temperature, which has nothing to do with aggregate but is only caused by the temperature sensitivity of asphalt. Figure 9 shows the relationship between $\omega_c$ and the phase angle of the asphalt binder at the same temperature, with $R^2$ greater than 95%, which is sufficient to illustrate this temperature sensitivity issue.

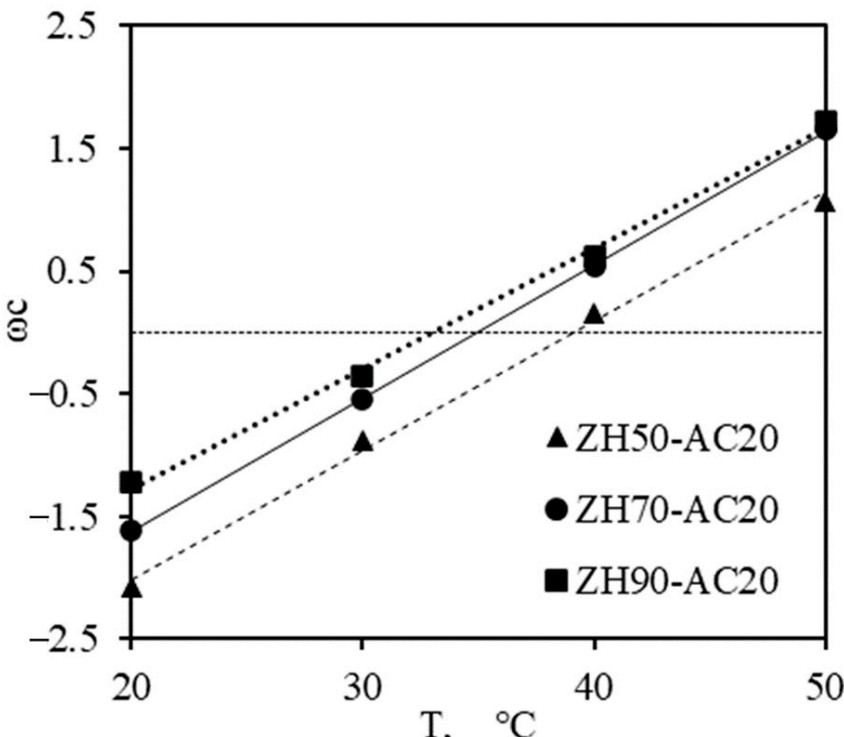

**Figure 8.** Relationship between $\omega_c$ and temperature.

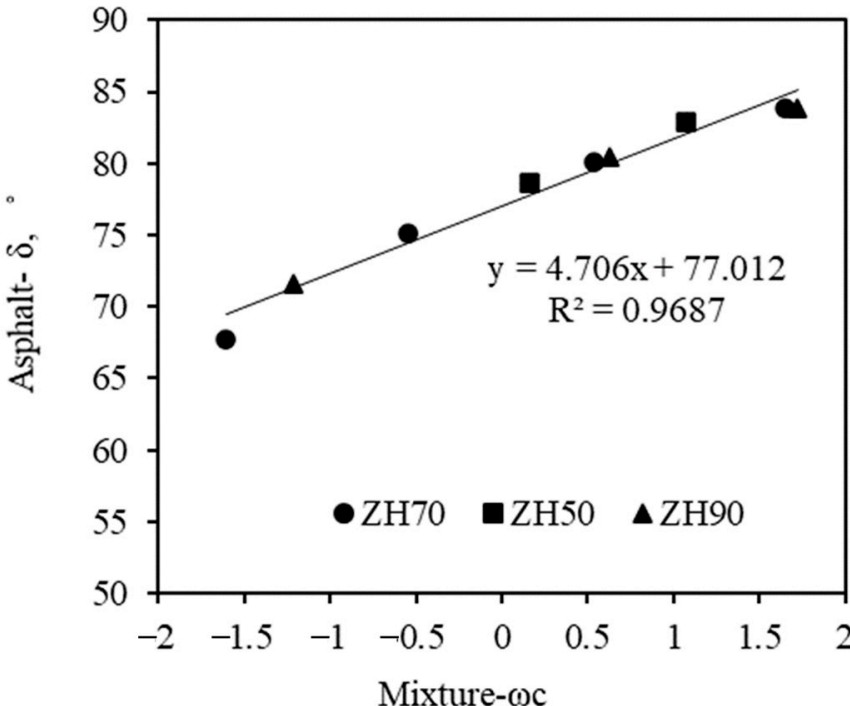

**Figure 9.** Relationship between $\omega_c$ and asphalt phase angle.

In the theory of polymer physics, the $\omega_c$ position is related to the relaxation characteristics of polymer materials, which is defined as the relaxation peak [10]. Its significance is to characterize the response ability of the viscoelasticity of materials under external action. Only when the load time (t) is close to the material relaxation time ($\tau$), that is, the action frequency ($\omega_r$) is the same as the reciprocal of the relaxation time ($\tau^{-1}$) ($\omega_r = \tau^{-1}$), is the energy consumption of the material the highest, which reflects the peak value of the phase

angle or loss angle tangent; when the frequency is high, the action time of external force is short, and the viscous pot effect of the material is too late to respond, reflecting the elastic characteristics and glass state. When the frequency is low, the action time of the external force is long, the spring effect of the material is completely restored, which shows the full sticky pot effect, and the material shows more viscous characteristics and a rubber state.

Because the asphalt mixture will not appear with absolute flow behavior, there will not be multiple relaxation characteristic peaks like polymer materials, and there is only one phase angle peak or loss angle tangent peak. Therefore, at a certain temperature, the peak position of the phase angle of the asphalt mixture can be used as a reflection of its relaxation ability, which is also important information given by the phase angle master curves, which is often ignored by researchers. According to the peak position of the phase angle master curve, the relaxation time and ability of the mixture at a certain temperature can be calculated and investigated according to the relationship $\omega = \tau^{-1}$. The relaxation times of three kinds of asphalt mixtures at different temperatures are shown in Table 3.

**Table 3.** Peak position of phase angle and relaxation time.

| Mixture Type | Temperature, °C | $\omega_c$ | $10^{\omega_c}$, Hz | $\tau$, s |
|---|---|---|---|---|
| ZH50-AC20 | 20 | −2.084 | 0.008 | 121.34 |
| | 30 | −0.889 | 0.129 | 7.74 |
| | 40 | 0.161 | 1.449 | 0.69 |
| | 50 | 1.074 | 11.858 | 0.08 |
| ZH70-AC20 | 20 | −1.607 | 0.025 | 40.46 |
| | 30 | −0.551 | 0.281 | 3.56 |
| | 40 | 0.536 | 3.436 | 0.29 |
| | 50 | 1.654 | 45.082 | 0.02 |
| ZH90-AC20 | 20 | −1.216 | 0.061 | 16.44 |
| | 30 | −0.364 | 0.433 | 2.31 |
| | 40 | 0.624 | 4.207 | 0.24 |
| | 50 | 1.725 | 53.088 | 0.02 |

Figure 10 shows the relationship between relaxation time and temperature. It shows that in the range of 20 to 50 °C, the higher the temperature is, the smaller the relaxation time of the asphalt mixture is, and it shows an exponential attenuation relationship. For penetration grading 50 asphalt, the relaxation time of AC20 asphalt mixture is 121.3 s at 20 °C, but it can be reduced to 0.08 s at 50 °C. This explains the reason for the cracking disease caused by the stress concentration of the asphalt mixture under the condition of low temperatures. In addition, for the same kind of asphalt mixture, the high-grade asphalt has less relaxation time; for example, at 20 °C, the relaxation times of grades 50, 70, and 90 asphalt mixtures are 121.3, 40.5, and 16.4 s, respectively. This shows that high-grade asphalt is helpful for improving the cracking resistance of the mixture. At the same time, the relaxation time is of great significance to the characterization of crack resistance of the asphalt mixture. Under the condition of low temperatures, cooling will bring greater temperature stress; if the relaxation time of the asphalt mixture is short, it can relax most of the temperature stress and greatly reduce the possibility of pavement cracking. This is consistent with the conclusion of the existing study using the relaxation time spectrum [12,13].

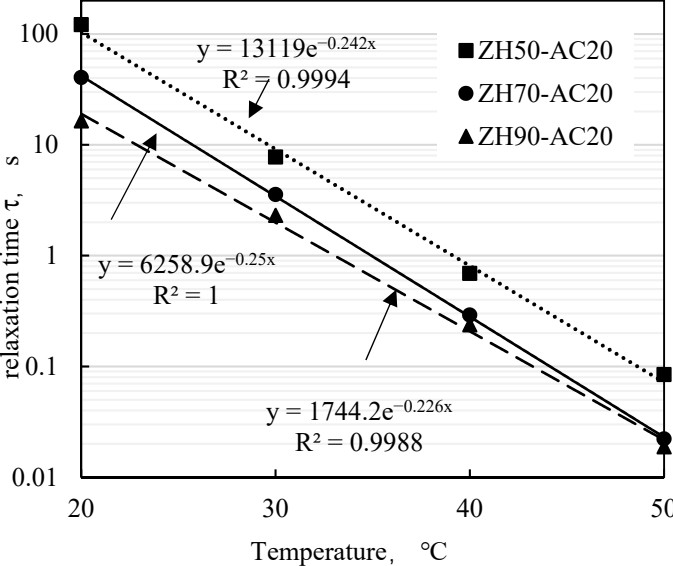

**Figure 10.** Relationship between relaxation time (τ) and temperature.

## 4. Master Curves of Different Asphalt Mixture

Table 4 shows the material types and volume indexes of several kinds of asphalt mixtures. The data of the dynamic modulus and phase angle were obtained on an AMPT tester. The master curve parameters were analyzed by Equations (1) and (2), and the main curve diagram of the mixture at 40 °C was drawn.

**Table 4.** Material types and volume indices of several asphalt mixtures.

| Code | Asphalt Type | Mixture Type | Pb, % | Vv, % | VMA, % | VFA, % |
|------|-------------|-------------|-------|-------|--------|--------|
| ZH50-AC20 | Pen-grade 50 | HMA, AC20 | 4.3 | 4.6 | 14.2 | 67.32 |
| ZH70-AC20 | Pen-grade 70 | HMA, AC20 | 4.4 | 4.4 | 14.1 | 68.54 |
| ZH90-AC20 | Pen-grade 90 | HMA, AC20 | 4.3 | 4.4 | 14.4 | 68.47 |
| SBS-SMA13 | SBS modified asphalt | HMA, SMA13 | 6 | 4.2 | 17.1 | 75.6 |
| MAC-AC20 | MAC modified asphalt | HMA, AC20 | 4.4 | 4.3 | 13.4 | 68.3 |
| ZH70-AC-25 | Pen-grade 70 | HMA, AC25 | 4.1 | 4.1 | 12.6 | 68 |
| MAC-LSPM30 | MAC modified asphalt | HMA, LSPM30 | 3.2 | 15.2 | 23.5 | / |
| SBS-AC13 | SBS modified asphalt | HMA, AC13 | 6.2 | 2.4 | 15.2 | 82.9 |
| E-CRM | Emulsified asphalt | Cold recycling | 3.5 (plus 1.5%cement) | 10.2 | / | / |
| F-CRM1 | Formed asphalt | Cold recycling | 3 (plus 1.5%cement) | 11.5 | / | / |
| F-CRM2 | Formed asphalt | Cold recycling | 3 (plus 1.5%cement) | 12.0 | / | / |

(1) Figure 11A shows the modulus master curve of the asphalt mixtures. Due to the difference in aggregate gradation, asphalt content, and volume index of asphalt mixtures, the difference in the dynamic modulus is obvious; due to the different form and properties of the asphalt binder, the shape of the modulus master curve is obviously different, and the shape parameters (β) are also different. Figure 12 shows that grade 50, grade 70, and grade 90 petroleum asphalt is obviously different from SBS and MAC-modified asphalt, and the modulus is more sensitive to frequency. At the same time, it is also different from the binder form of emulsified asphalt (containing cement) and foamed asphalt (containing cement). The existence of emulsified asphalt and foamed asphalt combined with cement can improve the low-frequency ultimate modulus of the mixture. The dynamic characteristics of different asphalt mixtures are determined by the factors of mixture and binder.

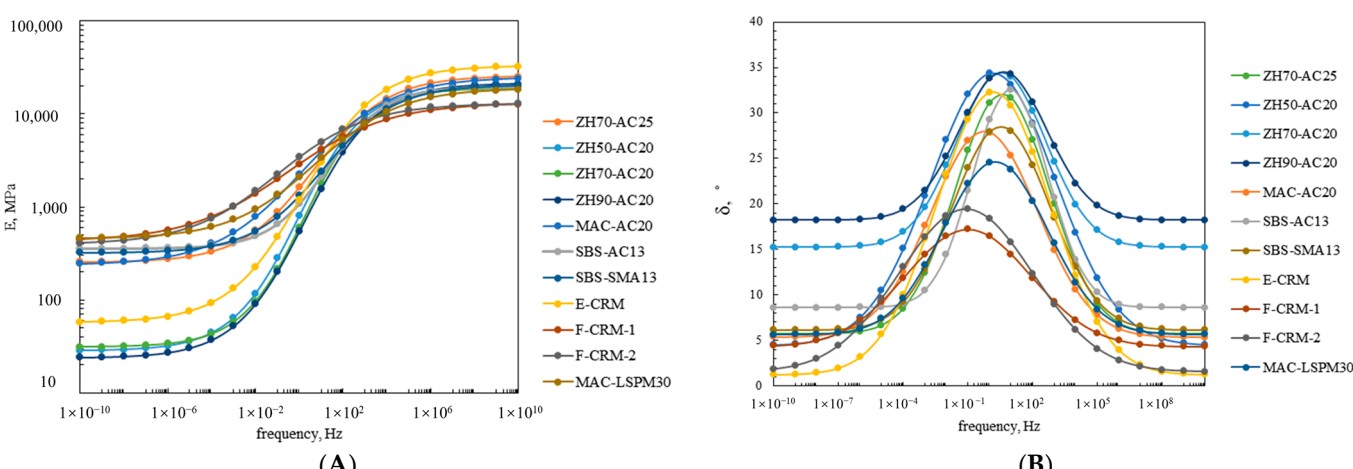

**Figure 11.** Comparison of master curves of different asphalt mixtures. (**A**) Dynamic modulus master curve, (**B**) Phase angle master curve.

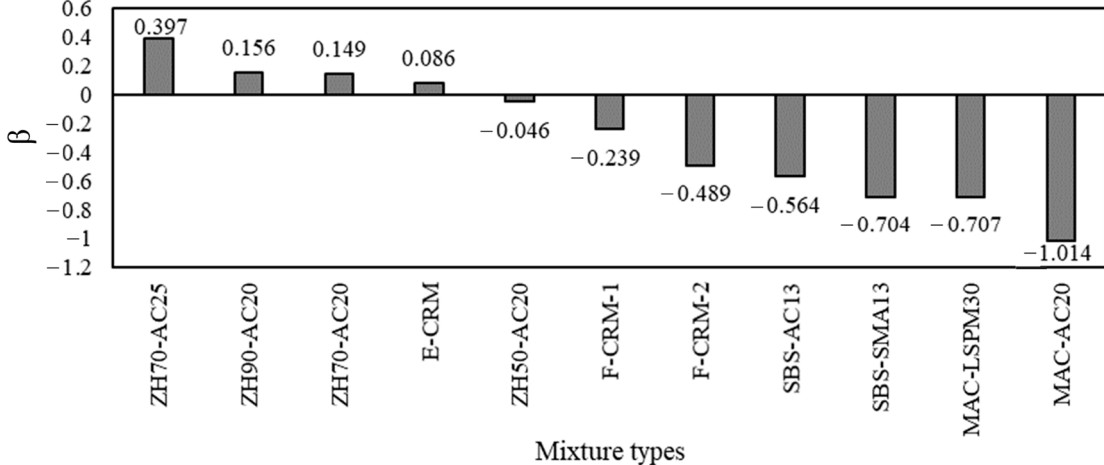

**Figure 12.** Comparison of shape parameters (β) of modulus master curves.

(2) Figure 11B shows the phase angle master curve of the mixture. Table 5 shows the phase angle master curve parameters. Due to the different forms and properties of the binder, the phase angle master curve of the mixture is obviously different. Among them, the change range of phase angle (A) of foam cold-recycled asphalt mixture is much smaller than that of other asphalt mixtures and emulsified asphalt mixtures. Figure 13 shows that its A/ω value is the smallest, reflecting the weakest viscoelastic characteristics, and tends to be rigid, which is related to the cement cementitious state and the non-continuous distribution of foamed asphalt on the aggregate surface. Although the cold-recycled mixture of emulsified asphalt has a lower initial phase angle ($\delta_0$), it shows more significant viscoelasticity and the variation range of FWHM (ω) and phase angle height (A) is larger than that of foamed asphalt. This is related to the continuous distribution of emulsified asphalt among aggregate particles, shown in Figure 14. For the binder of the hot-mix asphalt mixture, SBS-modified asphalt mixture SBS-AC13 has the best viscoelastic performance, and its A/ω ratio is the highest.

**Table 5.** Fitting parameters of phase angle master curves.

| Mixture Type | $\delta_0$, ° | A, ° | $\omega_c$, Hz | $\omega$, Hz | A/$\omega$ |
|:---:|:---:|:---:|:---:|:---:|:---:|
| F-CRM-2 | 1.53 | 17.94 | 0.08 | 1233.81 | 0.01 |
| F-CRM-1 | 4.29 | 12.92 | 0.10 | 800.49 | 0.02 |
| MAC-AC20 | 5.35 | 22.63 | 0.57 | 303.40 | 0.07 |
| ZH50-AC20 | 4.44 | 29.96 | 1.45 | 781.38 | 0.04 |
| E-CRM | 1.20 | 31.10 | 1.53 | 432.83 | 0.07 |
| MAC-LSPM30 | 5.64 | 19.05 | 1.86 | 259.68 | 0.07 |
| SBS-SMA13 | 6.16 | 22.34 | 3.37 | 187.31 | 0.12 |
| ZH70-AC20 | 15.26 | 19.25 | 3.44 | 115.51 | 0.17 |
| ZH70-AC25 | 5.70 | 26.40 | 3.92 | 148.42 | 0.18 |
| ZH90-AC20 | 18.22 | 16.31 | 4.21 | 105.94 | 0.15 |
| SBS-AC13 | 8.62 | 23.98 | 8.96 | 56.52 | 0.42 |

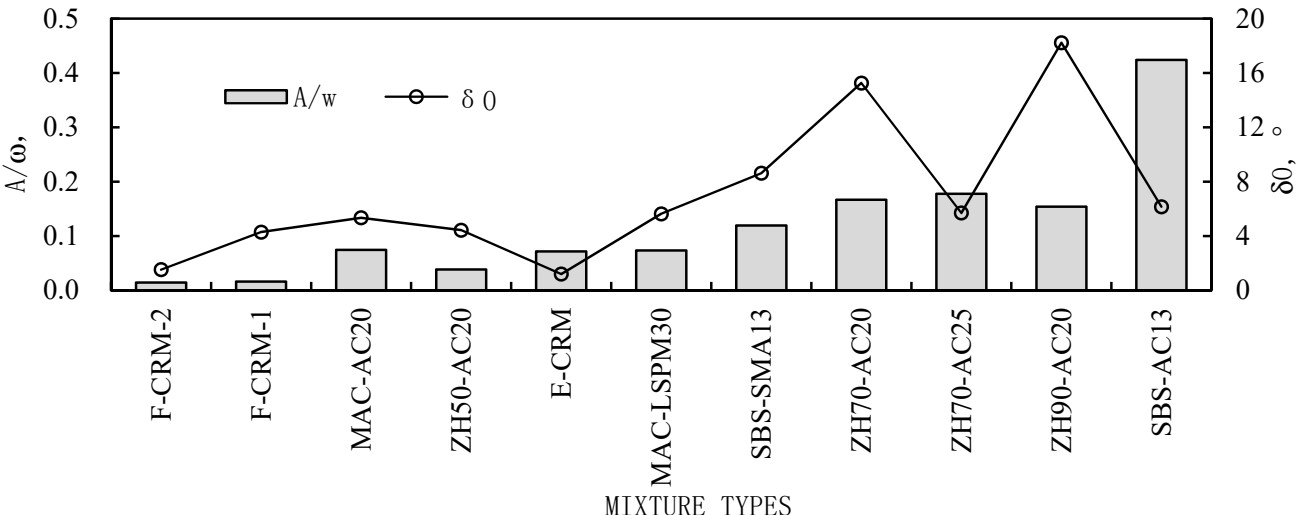

**Figure 13.** Comparison of master curve parameters (A/$\omega$, $\delta_0$) of phase angle.

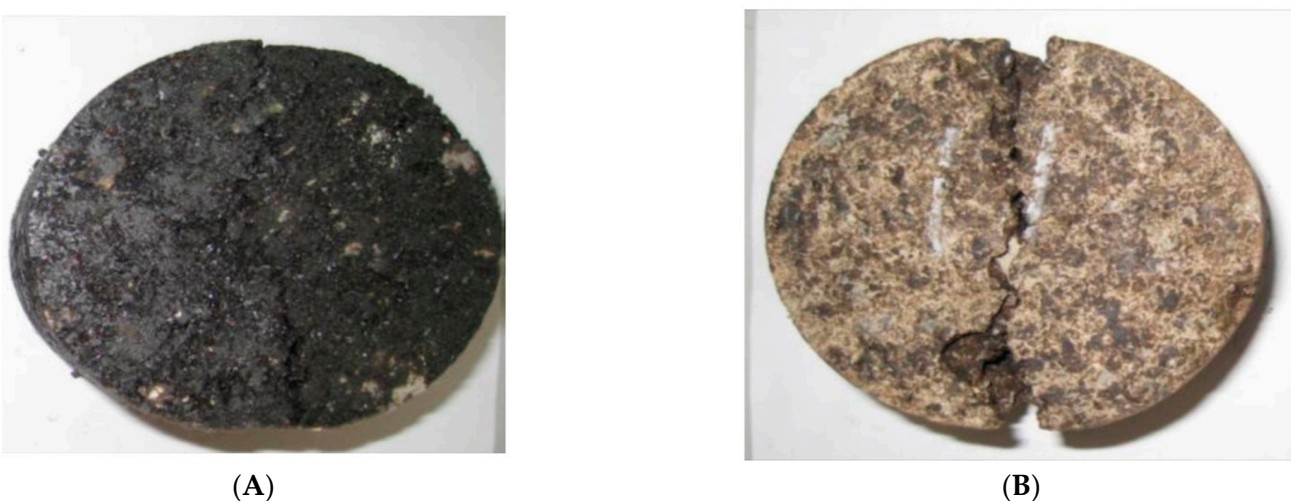

(**A**)          (**B**)

**Figure 14.** Cold recycling asphalt mixture (**A**) Emulsified asphalt cold recycling, (**B**) Foam cold recycling.

(3) Due to the different phase compositions and viscoelasticities of different asphalt mixtures, they reflect different stress relaxation behavior and relaxation times under specific temperatures and alternating stress environments. According to the phase angle master curve parameter $\omega_c$, the relaxation time of different mixtures can be calculated according

to the same calculation method in Table 2. Figure 15 shows the comparison of relaxation time of different asphalt mixtures. Among them, the relaxation time of foam cold-recycled mixture is the longest, which is significantly higher than that of other forms of asphalt mixture; the relaxation time of E-CRM cold-recycled mixture is similar to that of AC20 and LSPM30 mixtures; SBS-modified asphalt mixture SBS-AC13 has the best relaxation characteristics. In the current cold-recycling technology for a flexible base, it can be predicted that foam cold-recycling cannot achieve the effect of the LSPM30 mixture, but emulsified asphalt cold-recycling can.

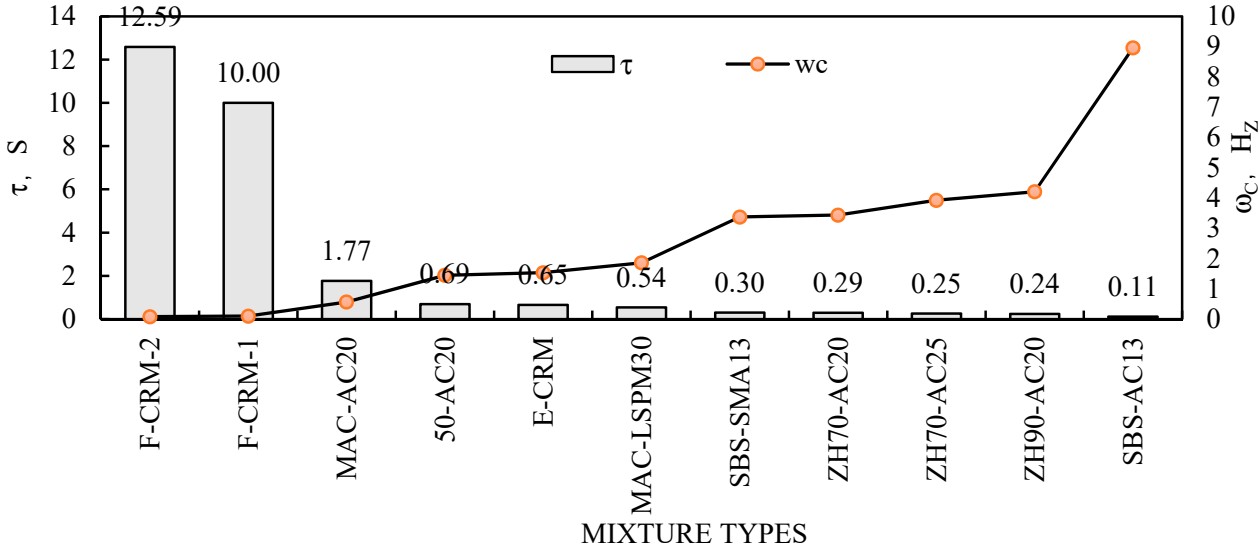

**Figure 15.** Comparison of relaxation time of different asphalt mixtures.

In this sense, the peak position parameter $\omega_c$ of the phase angle master curve can evaluate the relaxation characteristics of the mixture.

## 5. Conclusions

(1) The modulus master curve parameters ($E_0$, $\alpha$, $\gamma$) of asphalt mixtures are closely related to the mineral voidage (VMA), mixture density ($\gamma_f$), and asphalt content (Pb%) of the asphalt mixture. For the same kind of asphalt mixture, because the gradation of mineral aggregate is fixed, the volume parameters are almost the same, and the ultimate modulus of the mixture at different temperatures is unique. When the temperature changes or the asphalt changes, the shape parameter $\beta$ of the modulus master curve changes regularly, which brings different dynamic responses, and the lower $\beta$ will show higher modulus characteristics.

(2) Asphalt is the source of the viscoelasticity of the asphalt mixture. Although the influence of particle gradation of the mixture will bring about the change of modulus, the phase angle of the mixture depends on the viscoelastic properties of the asphalt. The initial phase angle ($\delta_0$) in the main curve is positively correlated with asphalt penetration and negatively correlated with softening point and viscosity, the peak height A is negatively correlated with penetration and softening point viscosity, and the viscoelastic interval represented by FWHM ($\omega$) is negatively correlated with penetration, but positively correlated with the softening point and viscosity.

(3) The peak position parameter $\omega_c$ of the phase angle master curve can evaluate the relaxation characteristics of the mixture and avoid the complex process of model calculation. In the comparison of the relaxation time of the asphalt mixture, the relaxation time of the foam cold-recycled mixture is the largest, which is significantly higher than that of other forms of asphalt mixture; emulsified asphalt cold-recycled mixture is equivalent to AC20 and LSPM30 mixtures; SBS-modified asphalt mixture

has the best relaxation characteristics; different relaxation times indicate different crack resistance, the longer the relaxation time, the worse the crack resistance. However, less relaxation time (less viscosity) leads to a large viscous flow in a very short time, creating a large rutting, which is one of the most serious issues in asphalt pavement on heavy-duty roadways.

**Author Contributions:** Investigation, S.M. and T.M.; Methodology, L.F. and Y.Z.; Experiment, Z.D.; Writing—original draft, S.M. and L.F.; Writing—review and editing, X.Z. All authors have read and agreed to the published version of the manuscript.

**Funding:** This research was funded by the Applied Basic Research Project of the Ministry of Transport, grant number 2013319781140, and project ZR2020QE271 supported by Shandong Provincial Natural Science.

**Institutional Review Board Statement:** Not applicable.

**Informed Consent Statement:** Not applicable.

**Data Availability Statement:** Not applicable.

**Conflicts of Interest:** The authors declare no conflict of interest.

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
