# Peer review of "Dynamic Characteristic Master Curve and Parameters of Different Asphalt Mixtures"

_applsci, doi:10.3390/app12073619_

Round 1

Reviewer 1 Report

Presented paper analyses influence of different bitumen types and different asphalt mixtures on parameters of dynamic modulus master curve and phase angel master curve. The analysis performed is very comprehensive and includes different correlations and relationships.

Some specific comments:

Section 2, line 94, the abbreviation “FWHM” was used for first time, so it is recommended to write the full term (I suppose this is Full Width Half Maximum).

Section 3.2.1, figure 7., new mixtures are introduced (SMA-13, LSPM-25, AC-13F, AC-25) and from the text it is not clear are these mixtures tested in this research or is the data from some previous studies. Please elaborate!

Section 4, line 271-272, it is stated: ”SBS modified asphalt has the best viscoelastic performance, and its A/ω ratio is the highest“, this is true only for SBS-AC20 mixture, and not SBS-SMA13; Please explain and if necessary rephrase the sentence.  

On some places in the text mixture LSPM20 (line 284, 315) is mentioned and in all figures and tables mixture LSPM30 is presented; Correct the wrong mixture designation!

Manuscript should be proofread.

Reviewer 2 Report

This paper presents experimental results of study on dynamic characteristics of asphalt mixtures expressed with master curves of dynamic modulus and phase angle. The authors adopted mathematical functions to approximate the curves and obtained the parameters required for the function based on the AMPT tests results conducted at different temperature and load frequencies. Also, they found surprisingly good correlation between the obtained parameters and mixture indexes that are commonly used in the mixture design. Although the regression process is not clearly explained, the obtained results are reasonable and corresponding well to the previous similar studies and pavement engineer’s experiences. The correlations between mater curve parameters and mixture index could be very useful if it is able to directly relate the mixture design to the structural model to predict the total performance of an asphalt pavement. In this sense, this paper provides valuable aspect to characterize the dynamic performance of asphalt mixtures and predict dynamic mechanical behavior of the asphalt pavement structure.

The reviewer has the following comments to be considered for revision.

Line 84. What does “V” mean ? It does not appear in Equation (2).

Line 105. How did you determine the mixture proportion of the asphalt mixtures in Table 1. The asphalt contents of the mixtures and aggregate composition were determined at the optimal ones? Otherwise, it was determined following the specification of China?

Line 113, Figure 2. The measured values of modulus and phase angle should be plotted in Figure 1 to verify the validity of the equations derived. Also, you should present the coefficient of determination of the regression analysis.

Line 129, the correlation analysis. The variation in the indexes of the asphalt and asphalt mixtures in Table 1 are quite small, less than a couple of %. In that case, do the correlation coefficients in Figure 3 have a significant meaning?

Line 134, Figure 3. Delta in the legend must be E0.

Line 196, Figure 8. The parameter “wc” is defined as the frequency corresponding to the peak of the phase angle in Equation (2). If so, why is it has a negative value? What the negative frequency mean?

Line 217. What does “(relaxation?) ability of the mixture at a certain temperature” mean? Does it mean that the stress in the mixture will decrease very rapidly?

Line 220, Table 3. The relationship between third and fourth columns is not correct. You can not calculate the fourth column value by 10 * wc.

Line 230. Too low viscosity leads to a large deformation and large rutting of asphalt pavement, which is a big problem. The authors should make a comment on the issue.

Line 270. “this is related to the continuous distribution of emulsified asphalt among aggregate particles.” Do you have an evidence or reference confirming the statement?

Line 279. “w” is missing just before “c”.

Line 284. What does “the best relaxation characteristics” mean specifically? For what is the best? Does it mean the best for crack resistance or rutting resistance?

Line 285. This sentence is not understandable. What does “the effect of LSPM” mean specifically?

Line 291. Please explain “the complex process of model calculation” more specifically. What kind of the calculation do you mean?

Line 315. “LSPM20” must be “LSPM30.”

Line 316. “the longer the relaxation time, the worse the crack resistance.” However less relaxation time (less viscosity) leads a large viscous flow with a very short time creating a large rutting, which is one of the most serious issues in asphalt pavement on heavy duty roadways. The authors should mention the issue.         
